# Political Stability and Bank Flows: New Evidence

## Mafalda Venâncio de Vasconcelos

Department of Economics and Statistics, Università degli Studi di Torino, 10134 Torino, Italy; avenanci@unito.it
or amvasconcelos@netcabo.pt

**Abstract:** In this paper, we use a rich dataset of several countries to analyze how sound political measures affect cross-border bank flows. Furthermore, our work is the first to comprehensively examine various components of political stability on the aforementioned subject using a larger sample than previous studies, and covering the period 1984–2013. Our paper will inform policy makers which particular aspects of political stability have a significant effect on cross-border bank flows and provide an outline on the favorable long term political and institutional development to increase such flows. We find that sound political measures—and therefore, higher political stability—increase cross-border bank flows, especially in advanced economies. Moreover, we find that in advanced economies, the political stability components; socioeconomic conditions, investment profile, corruption within the political system, religious tensions, ethnic tensions, and bureaucracy quality have a positive and close association with such bank flows. In our work, we also find that policies aiming to increase political stability have a stronger impact after the financial crisis of 2008, namely with regard to policies that affect socioeconomic conditions, investment profile, corruption within the political system and religious tensions.

**Keywords:** cross-border bank flows; political stability; economic risk; financial risk

## 1. Introduction

Over the past decades, the world has witnessed a global market increase in cross-border bank flows. After years of growth and increase in financial globalization, international banking activity suffered a sharp reduction with the financial crisis of 2007–2009. The retrenchment of international banking activity was highly pronounced, especially in advanced economies, with banks severely scaling back their operations. Developing markets were also hit by a reversal of flows, although the impact was not so relevant. Despite this reversal of trends, and according to the statistics of the Bank for International Settlements, global cross-border claims rose by $134 billion during the last quarter of 2018 and stand at $29 trillion at the end of this year. Understanding the drivers of cross-border bank flows has become a very important concern for policymakers, given its vital role in supporting economic development, promoting prosperity and reducing poverty. Moreover, international banking activity promotes gains in terms of required capital, know-how and technological improvements that foreign banks can bring, and that can lead to a more competitive and diversified banking system.

Several studies have analyzed the role of external (or push) and domestic (or pull) factors underlying bank flows and arrived at different conclusions with respect to their relative importance.

In the group of external factors, several drivers have been found to have an effect on the behavior of cross-border bank flows. For example, Herrmann and Mihaljek (2013) found that global risk aversion and expected financial market volatility were important in explaining the decrease in cross-border bank flows observed during the financial crisis that started in 2007. The role of global risk aversion

was also highlighted in the work by Forbes and Warnock (2012). Milesi-Ferreti and Tille (2011) found that the retrenchment in international capital flows during the financial crisis was due to credit risk aversion. While some works found the importance of external factors in driving international bank flows, others found that both external and domestic factors might have equal importance in explaining these flows. With regard to internal factors, works propose several important drivers of capital movements. Hernandez and Rudolph (1995), for example, found that domestic factors that reflect a country's investment attractiveness (domestic investment and saving rate, and export growth) are important in explaining total private capital flows (composed by foreign direct investment (FDI), portfolio and loans). Another work that analyzes the importance of internal factors was the one by, Choi and Furceri (2019). In this study, they argue that higher uncertainty (measured by country-specific stock market volatility) reduces cross-border bank flows from/into this country.

Political stability and the vulnerability of governments and their laws have affected all sectors of the economy, with consequences that are visible not only at the micro, but also at the macro level. The IMF and the World Bank, for example, have constantly urged their members to implement institutional reforms, tackle corruption, improve their bureaucracy and privatize state-owned companies to attract foreign capital. One of the most relevant fields that has been affected by changes in political stability is the banking sector, with ongoing news in the media that report information about the state of the aforementioned sector and how is it shaping a new paradigm of the world economy. Therefore, in the last decade, the role of political stability as a driver of international capital flows has become an important field of research. Although there is extensive literature on the determinants of international capital movements research, these, to a large extent, ignore the role of political risk on international banking flows. The focus has been mainly on the analysis of foreign direct investment. Considering that this type of capital is expected to be more stable and slanting towards reversals that bank flows, we expect the fundamentals that rely on foreign investors' behavior to be somewhat similar between these two types of international capital flows. A relevant work that explores the aforementioned issue was the one by Wei (2000), who found a negative association between corruption within the political system and FDI inflows. Other work that was in the same area of research was the one by Gastanaga et al. (1998), who found that lower corruption and nationalization risk levels, and better contract enforcement, are associated with higher FDI inflows. Several other studies analyze the effect of different types of political components on FDI flows. One example was the work by Busse and Hefeker (2005), who found that government stability, the absence of internal conflicts and ethnic tensions, the existence of basic democratic rights and the insurance of law and order are important determinants of such a type of capital flow.

Another relevant study related to this issue was the one by La Porta et al. (2002), who gave an important contribution to the role of politics on the banking system. They found that when state banks dominate the banking system, there is less interest in overseas banks, since the interest on political goals prevails over profit maximization. Another relevant work was the one by Papaioannou (2009), who showed that poorly performing institutions, such as weak protection rights, legal system inefficiency and a high risk of expropriation, were significant impediments to foreign bank capital.

The importance of international bank business as a vital industry to economic growth makes the analysis of drivers of bank flows a very important field of research that needs to be explored. In this section, we highlighted the literature that provides—to the best of our knowledge—the best explanation to the drivers of international bank business, namely the ones related to economic and financial issues. We further contributed to the literature by providing a deeper comprehensive analysis on the role of political stability in driving international bank flows.

The remainder of the paper is structured as follows: the next section describes the methodology, data and descriptive statistics and other evidence. Section 3 features the estimation results. This is the section where, broadly defined, the effect of political stability on cross-border bank flows is analyzed. This analysis further extends to both OECD and non-OECD countries and to before and after the financial crisis. In this section, we also present an analysis of the impact of specific political stability

components on cross-border bank flows in OECD countries, before and after the financial crisis. Conclusions are then presented in Section 4.

## 2. Methodology, Data, Descriptive Statistics and Other Evidence

### 2.1. Methodology

The econometric work is carried out using a fixed effects model, with clustered standard errors, and according to the following specification:

$$\log Y_{i,t} = \beta_0 + \beta_1 X_{1i,t-1} + \beta_2 X_{2i,t-1} + \beta_3 X_{3i,t-1} + \beta_4 X_{4,t-1} + \alpha_i + \alpha_t + e_{i,t}$$

where $i$ and $t$ indicate country and time (quarter), respectively. The dependent variable, $\log Y_{i,t}$, measures the logarithm of cross-border bank flows reported by all countries to a counterparty country $i$. The $X_1$ variable measures the level of political stability in a country; $X_2$ reports the GDP volume (% change); $X_3$ is the economic risk; $X_4$ is the financial risk; $\alpha_i$ and $\alpha_t$ present country and time fixed effects, respectively; and $\varepsilon_{i,t}$ accounts for the error term.

On one hand, controlling for country fixed effects accounts for (to a first-approximation time invariant) social norms, culture, geography and trust, which affect both finance and institutional quality. On the other hand, controlling for time fixed effects enables us to capture the influence of aggregate (time-series) trends.

### 2.2. Data

We perform our empirical analysis with a combination of several data sources, covering the period between the second quarter of 1984 and the last quarter of 2013.

The literature suggests several variables as potential drivers of bank flows and the different studies do not seem to agree on a fixed set of determinants. In our work, we follow other researchers and we choose the variables that are the most common in the literature.

With regard to cross-border bank flows, we use data taken from the Locational Banking Statistics of the Bank for International Settlements (BIS). This dataset comprises quarterly data on international banking business conducted in countries and other financial centers. This type of information is interesting because it is in accordance with the balance of payments and external debt methodology. Therefore, using this type of data, we can better understand the role of bank flows in shaping the stance of the balance of payments of a country in a particular period.

In our study, we use data on cross-border bank claims reported by all countries (reporting countries) to a counterparty country $i$ (a detailed description of the countries included in our sample is reported in Table A1 in the Appendix B). Claims are composed by loans and deposits (that comprise interbank deposits and loans and advances to bank and non-banks), holdings of securities and other claims (namely, equity securities, participations, derivatives, instruments with positive market value and other residuals on balance sheet financial claims). Reporting institutions include domestic and foreign banks, consortium, and others that are unclassified (for example, banks that cannot be classified according to a single controlling parent institution). With regard to the counterparty sector, it includes banks (related offices and central banks), non-bank financial institutions, non-financial sectors (general government, non-financial corporations and households including non-profit institutions serving households). In reference to its measure, we use the adjusted changes since it gives us an approximation of the flow between two points in time. According to the Balance of Payments and International Investment Position Manual, IMF, Sixth Edition (2009), a claim is a financial instrument that gives rise to an economic asset that has a counterparty liability. Claims arise from contractual relationships entered into when one institutional unit promises to provide funds or other resources to another in the future. Each claim is a financial asset that has a corresponding liability. The existence of two parties to a claim means that it can arise in a cross-border situation.

Therefore, positive asset flow means that the capital that is leaving the reporting country on net by domestic residents (and entering the counterparty country *i*) increases from period *t* to period *t+1*. On the other hand, negative asset flows mean that, from period *t* to period *t+1*, the capital that is leaving the reporting country decreases (and enters the counterparty country *i*) from period *t* to *t+1*.

With regard to the calculus of the aforementioned variable, and since cross-border bank flows can take negative values, we first calculated the logarithm of the absolute value and then we assigned to it a negative sign. Using this functional form enabled us to interpret our coefficients as semi-elasticities.

We also collected data at a quarterly frequency of the GDP volume (% change) from the International Financial Statistics of the International Monetary Fund (line 99).

Other data is retrieved from the International Country Risk Guide (ICRG). This dataset comprises several variables related to categories of risk. It comprises variables (and its subcomponents) related to political, financial and economic risk in countries at a monthly frequency.

Our variable of interest, according to the ICRG nomenclature, is the political risk rating, that we will mention in our paper as "political stability". The aim of this variable is to provide a way to assess the political stability in countries, giving them risk points. The aforementioned variable is based on 100 points, where higher values mean lower political risk, and therefore, higher political stability in a country. This index includes 12 weight variables (where higher values are given to lower risk and vice-versa) that represent political and social attributes of a country at a certain point in time, namely: government stability; socioeconomic conditions; investment profile; internal conflict; external conflict; corruption; military in politics; religious tensions; law and order; ethnic tensions and democratic accountability. We use these variables in a latter section, where we explore the political stability features that may drive cross-border bank flows. A detailed description of the aforementioned variables is given in the Appendix A. Another variable that we use in our study is the economic risk rating, here mentioned as "economic risk". It assesses the economic strength and weakness of a country. This variable is also measured in points, where higher values are given to countries with better economic performance. It is determined by a weight group of economic risk components, namely: GDP per head; real GDP growth; annual inflation rate; budget balance as a percentage of GDP and current account as a percentage of GDP. The country´s ability to finance its official, commercial and trade obligations is given by the financial risk rating, mentioned in our work as "financial risk". This variable is also measured in points, where higher values are given to lower financial risk. In other words, higher values are given to countries that have higher chances to pay what they own, and therefore, less likely to face a sudden financial crisis. This variable is composed by several weighted components, namely: foreign debt as a percentage of GDP; foreign debt service as a percentage of exports of goods and services; current account as a percentage of exports of goods and services; net international liquidity as months of import cover, and exchange rate stability.

Given the fact that political, economic and financial risk rating variables are given at a monthly frequency, we calculate the mean by quarter to achieve the quarterly frequency required to our analysis.

Moreover, the components that comprise our risk variables are embracing and therefore, give us a close measure of the overall political, economic and financial perspective of a country.

Hence, we can clearly assume that we include in our model all relevant variables that can accurately determine cross-border bank flows.

*2.3. Descriptive Statistics and Other Evidence*

In this work, we use a sample of 71 countries (34 OECD and 37 non-OECD countries) and a time period between the second quarter of 1984 and the last quarter of 2013.

In Tables A1–A3 in the Appendix B, we summarize the descriptive statistics of the variables employed in the analysis using all the countries, OECD countries and non-OECD countries, respectively.

Descriptive statistics show that cross-border bank flows differ considerably, with a minimum of −339 billion dollars and a maximum of 441 billion dollars (the logarithm of the aforementioned variable presents a minimum of −33.46 and a maximum of 33.72). To provide a first look at this data on

bank flows, we plot the cross-border bank flows (Figure A1) and political stability (Figure A2) for three countries (Germany, Brazil and Mexico), where we can clearly observe the dominance of Germany over Brazil and Mexico with regard to the volume and volatility of cross-border bank flows, reflecting that richer countries engage more in cross-border banks transactions. With reference to developed and non-developed economies, the statistics show that the mean of the aforementioned variables for OECD countries ($1.28 \times 10^{11}$) is higher than the one of non-OECD countries (467,646.5). This result suggests the possibility of the dominance of OECD countries over non-OECD countries with regard to the volume of international bank transactions. Moreover, we can clearly observe the relationship between political stability and cross-border bank flows, when we consider the case of the OECD country.

Through the statistics results we can also observe that political stability various enormously between countries with a minimum of 31.33 and a maximum of 97. The dispersion around the mean is also high as it is shown by its standard deviation. Moreover, in Figure A3, we plot the box-plot of political stability for OECD and non-OECD countries, where we can observe that, on average, OECD countries are less risky than non-OECD countries. Statistics related with other variables (namely mean, standard deviation, minimum and maximum) are also provided in the same table[1]. The association between the main variables used in this work are reported in the correlation matrix presented in Table A4.

## 3. Estimation Results

### 3.1. Main Results

Table A5 shows multiple estimation results. In columns one through three, we estimate the model using all countries; OECD countries and non-OECD countries, respectively. In columns four through five, we show the results obtained when we split the sample of OECD countries into before and after the financial crisis in 2008, respectively. In columns six through seven, we divide our sample of non-OECD countries into before and after the financial crisis, respectively.

We now turn to provide an explanation of the results.

In column one, we try to capture the influence of political stability on cross-border bank flows when controlling for GDP volume (% change), economic risk and financial risk. All coefficients consistently enter with well-behaved coefficients. The results confirm the importance of political stability as a driver of cross-border bank flows. The coefficient on this variable (0.235) is quite large and statistically significant at the one percent level, and implies that a 10 point increase in political stability increases international bank flows by almost 25 percent, conditioned on GDP volume (% change), economic risk and financial risk. The estimate is actually close to the results obtained in other works (i.e., Papaioannou) that find a positive and statistically significant coefficient for this variable that ranges between 0.26–0.28. The GDP volume (% change) enters with the largest coefficient (0.385) and with a statistical significance at the 1 percent level. Our intuition is that an increase in the economy's size is expected to increase the need for banks to finance its economic activity. The magnitude of the coefficient on economic risk reflects that good macroeconomic performance in a country attracts international bank lending. Since this variable covers measures such as the budget balance (as a percentage of GDP) or the current account balance (as a percentage of GDP), we can conclude that an improvement in the aforementioned items of the payments balance not only reflect an improvement in the wealth of an economy but also increases the chances of higher cross-border bank lending to a counterparty country. Concerning the financial risk, the positive coefficient (0.211), with a statistical significance of five percent, reflects what we were expecting if a country has higher changes to pay what it owns, as the chances to receive a loan from another country are higher.

---

[1]  Additional information may be given on request.

In most of the empirical analyses that study the effects of political stability, there is a concern as to whether the estimates are been driven by the substantial variability between rich and non-developed countries. Moreover, and since political stability differs considerably, on average, between OECD and non-OECD countries, we can infer that maybe its effect on cross-border bank flows is different when we consider either a developed or a non-developed economy. Given the aforementioned concerns, we re-estimated the model separately in OECD and non-OECD countries. Models two and three present the estimation results for each group of economies.

In column two, we restrict our analysis to OECD countries. While the number is small, since we have less countries in the estimation (34 countries), the coefficients are more precisely estimated. Using the sample of OECD countries, the results slightly change, and it seems that our model fits better when we consider only these countries (the R-squared increases from 0.152 to 0.198). In this re-estimation, the model can explain almost 20 percent of the variability in cross-border bank flows just with political stability, GDP volume (% change), economic risk and financial risk. Furthermore, all the coefficients retain the sign and the statistical significance when compared with the model estimated with the whole sample. Interestingly, the coefficient on political risk increases in magnitude, which reinforces our intuition that a political risk is an important driver of cross-border bank flows, especially in OECD countries. Policies targeting increased political stability in a country are expected to have positive repercussions on international bank lending. Since most of developed countries face a serious problem with deficits in the current balance and in the budget balance, an improvement in each of these aforementioned items is expected to be a sign of a recovery in the economy, and therefore more attractive to foreign investors. The increase in magnitude on the coefficient of the economic risk (from 0.258 to 0.319) reflects this issue. When we consider the sample of OECD countries, the coefficient on financial risk reflects the fact that the risk of the country not been able to pay what it owns is a matter of concern to the banking system, although the coefficient slightly decreased from 0.211 to 0.175 when we restricted our sample to developed economies. The coefficient on GDP volume (% change) is lower in magnitude (0.293) than the one we obtain when we use all the sample of countries, but it is still important to explain the response of cross-border bank flows.

In column 3, we restrict our sample to non-OECD countries, and we find evidence that political stability is of secondary importance to explain cross-border bank flows, and therefore, policies targeting increased political stability are not relevant to explain international bank business in this particular group of countries. The coefficient on GDP volume (% change) of 0.366 reflects the importance of wealth as a driver of cross-border bank flows. An improvement in the wealth of the nations is important to increase foreign lending to those countries. Investors are more willing to invest in a country if they expect profitable returns on their investments. The coefficient on financial risk (0.443) reflects that tackling financial risk in non-OECD countries is the most important factor if those countries decide to increase its cross-border bank activity. Developing policies that are able to improve the country's ability to financial its official, commercial and trade debt obligations can make foreign investors to feel more secure in investing in non-developed countries. On the other hand, economy risk seems to be of secondary importance in explaining cross-border bank flows in this restricted sample of countries.

Other Results—Financial Crisis

Given the importance of the financial crisis in shaping international bank business, in this section, we analyze how the aforementioned event has changed the perception of political stability as a driver of cross-border bank flows. To perform this analysis, we split our sample into OECD and non-OECD countries and we analyze the results, before and after the first quarter of 2008. The results of OECD countries before and after the event are shown in columns four through five, respectively. The ones of non-OECD countries are shown in columns six through seven, respectively.

The hypothesized results confirm the importance of political stability as a driver of cross-border bank flows, especially in OECD countries. Moreover, in advanced economies, policies targeting increased political stability have a higher impact on cross-border bank flows after the financial crisis.

The coefficients of political stability obtained in columns four and five confirm this last result, since it changes from 0.291 before the crisis to 0.718 after the crisis, and with a relevant statistical significance of 5 percent and 10 percent, respectively.

### 3.2. Political Stability Components

#### 3.2.1. Main Results

The analysis so far does not address which institutional features drive cross-border bank flows. While it is important for theory and policy advice how political stability behaves as a driver of cross-border bank flows, research has not explicitly tackled the mechanisms through which political stability influences investors' decisions. Therefore, in this section, we are going to detect which components of political stability are most significantly associated with cross-border bank flows. These components are some of the variables that were used to calculate the variable of political stability (namely, government stability, socioeconomic conditions, investment profile, corruption, religious tensions, law and order, ethnic tensions, democratic accountability, bureaucracy quality), and the notable association between these variables and political stability is illustrated in the correlation matrix in Table A4 in the Appendix B. Given the positive and close association between the aforementioned variables, we can clearly assume that several components manage to capture different aspects of the phenomenon investigated. Therefore, and to analyze this issue, we substitute the political stability measure by each of the components, and we analyze the results. Moreover, and since our aggregate measure of the political stability is not significant in our model when we restrict our sample to non-OECD countries, we only analyze its impact as a driver of cross-border bank flows in OECD countries.

In Table A6 in the Appendix B, we report the estimation results. In each column we add a different component of the political stability conditioned on GDP volume (% change), economic risk and financial risk. In all simulations, the coefficients on GDP volume (% change), economic risk and financial risk remain positive and quite stable at the one or five percent level (when we compare it with the model using the aggregate measure of political stability). With reference to our variable of interest, we find that among the components of political stability, the socioeconomic conditions, investment profile, corruption, religious tensions, ethnic tensions and bureaucracy accountability are closely related to cross-border bank flows. The coefficients on the aforementioned variables are positive and statistically significant at the one percent level, except ethnic tension (statistically significant at the 10 percent level) and bureaucracy quality (statistically significant at the five percent level). Our results suggest that sound policies that tackle unemployment and poverty and stimulate consumers' confidence increase the need for banks' financing, and therefore, we expect cross-border bank flows to increase. Furthermore, a strong investment profile is crucial to attract international bank flows to OECD countries. Some of the examples of good conditions for investment are the ones that are related to high chances of contract viability and profit repatriation and low chances of payment delays.

Furthermore, it seems that high corruption is an important impediment for cross-border bank flows. Corruption within the political system may deter investment, since it increases the cost of doing business and therefore distorts the economic and financial environment. An example of corruption may come in the form of special payments and bribes connected with tax assessments or loans. Moreover, it seems that foreign banks are unwilling to invest in countries with higher risks of religious tensions that may come by a possible domination of society (or governance) by a single religious group that desires to substitute civil law by religious law. Ethnic tensions also seem to be associated with cross-border bank flows, however the coefficient (0.926) is only statistically significant at the 10 percent level, and therefore we should be cautious in interpreting this result.

Our results suggest that the most important driver of international bank flows is bureaucracy quality. The perception that there are low revisions of policies when governments change substantially increases cross-border bank flows. The estimation results obtained in model nine expresses this

intuition. The coefficient of bureaucracy quality is positive and large in magnitude (2.357), although it is statistically significant at the five percent level. Government stability, law and order and democratic accountability seem to be of secondary importance in explaining cross-border bank flows in OECD countries.

### 3.2.2. Other Results—Financial Crisis

In this section, we analyze the effect of the components of political stability using OECD countries, before and after the financial crisis. The results are shown in Tables A7 and A8 in the Appendix B. We use the same procedure as in Section 3.2.1 and we add to the model, one by one, each component of political stability. Interestingly, the results confirm that policies targeting increased political stability are especially relevant after the financial crisis. Moreover, our estimations suggest that better socioeconomic conditions and investment profile, lower corruption within the political system and lower religious tensions are expected to increase cross-border bank flows after this event.

## 4. Conclusions

In this paper, we analyzed if sound political measures—and therefore, higher political stability—increase cross-border bank flows. We found that sound policies that increase political stability are important to increase cross-border bank flows—when controlling for GDP growth, economic risk and financial risk—especially in advanced economies.

Moreover, our estimates suggest that an increase in GDP growth, and better economic and financial environments, are important drivers of cross-border bank flows.

In our work, we also found that in advanced economies, better socioeconomic conditions and investment profile, lower corruption within the political system and lower religious and ethnic tensions, and higher bureaucracy quality are fundamental drivers of cross-border bank flows.

We also found that the recent financial crisis of 2008 had an important impact on the role of political stability has a driver of cross-border bank flows, namely in advanced economies. After the crisis, policies targeting increased political stability have a higher impact on cross-border bank flows, namely with regard to socioeconomic conditions, investment profile, corruption within the political system and religious tensions.

**Funding:** This research received no external funding.

**Conflicts of Interest:** The author declares no conflict of interest.

## Appendix A

### *Appendix A.1. List of Countries*

Argentina; Australia; Austria; Belarus; Belgium; Bolivia; Botswana; Brazil; Bulgaria; Canada; Chile; Colombia; Costa Rica; Cyprus; Czech Republic; Denmark; Ecuador; Egypt; El Salvador; Estonia; Finland; France; Germany; Greece; Guatemala; Hungary; Iceland; India; Indonesia; Iran; Ireland, Israel; Italy; Jamaica; Japan; Jordan; Kenya; Latvia; Lithuania; Luxembourg; Malaysia; Malta; Mexico; Mongolia; Morocco; the Netherlands; New Zealand; Norway; Panama; Paraguay; Peru; the Philippines; Poland; Portugal; Qatar; Romania; Russia; Saudi Arabia; Singapore; Slovenia; South Africa; Spain; Sri Lanka; Sweden; Switzerland; Thailand; Turkey; Ukraine; the United Kingdom; the United States; Uruguay.

### *Appendix A.2. Definition of the Political Stability Components*

**Government stability:** According to the ICRG, this is an assessment both of the government's ability to carry out its declared program(s), and its ability to stay in office. The risk rating assigned is the sum of three subcomponents, each with a maximum score of four points and a minimum score of 0 points. A score of 4 points equates to Very Low Risk and a score of 0 points to Very High Risk.

The subcomponents are:

- Government Unity
- Legislative Strength
- Popular Support

**Socioeconomic conditions:** According to the ICRG, this is an assessment of the socioeconomic pressures at work in society that could constrain government action or fuel social dissatisfaction. The risk rating assigned is the sum of three subcomponents, each with a maximum score of four points and a minimum score of 0 points. A score of 4 points equates to Very Low Risk and a score of 0 points to Very High Risk.
The subcomponents are:

- Unemployment
- Consumer Confidence
- Poverty

**Investment profile:** According to the ICRG, this is an assessment of factors affecting the risk to investment that are not covered by other political, economic and financial risk components. The risk rating assigned is the sum of three subcomponents, each with a maximum score of four points and a minimum score of 0 points. A score of 4 points equates to Very Low Risk and a score of 0 points to Very High Risk.
The subcomponents are:

- Contract Viability/Expropriation
- Profits Repatriation
- Payment Delays

**Corruption:** According to the ICRG methodology, corruption is an assessment of corruption within the political system. Such corruption is a threat to foreign investment for several reasons: it distorts the economic and financial environment; it reduces the efficiency of government and business by enabling people to assume positions of power through patronage rather than ability; and, last but not least, introduces an inherent instability into the political process. The most common form of corruption met directly by business is financial corruption in the form of demands for special payments and bribes connected with import and export licenses, exchange controls, tax assessments, police protection, or loans. Such corruption can make it difficult to conduct business effectively, an in some cases may force the withdrawal or withholding of an investment. Although our measure takes such corruption into account, it is more concerned with actual or potential corruption in the form of excessive patronage, nepotism, job reservations, "favor-favors", secret party funding, and suspiciously close ties between politics and business. In our view these insidious sorts of corruption are potentially of much greater risk to foreign business in that they can lead to popular discontent, unrealistic and inefficient controls on the state economy, and encourage the development of the black market. The greatest risk in such corruption is that at some time it will become so overweening, or some major scandal will be suddenly revealed, as to provoke a popular backlash, resulting in a fall or overthrow of the government, a major reorganizing or restructuring of the country´s political institutions, or, at worst, a breakdown in law and order, rendering the country ungovernable.

**Religious tensions:** this variable measures tensions among religious groups. According to the ICRG definition, religious tensions may stem from the domination of society and/or governance by a single religious group that seeks to replace civil law by religious law and to exclude other religions from the political and/or social process; the desire of a single religious group to dominate governance; the suppression of religious freedom; the desire of a religious group to express its own identity; separate from the country as a whole. The risk involved in these situations range from inexperienced people imposing inappropriate policies through civil dissent to civil war.

**Law and order:** According to the ICRG, "Law and Order" form a single component, but its two elements are assessed separately, with each element being scored from zero to three points. To assess the "Law" element, the strength and impartiality of the legal system are considered, while the "Order" element is an assessment of popular observance of the law. Thus, a country can enjoy a high rating—3—in terms of its judicial system, but a low rating—1—if it suffers from a very high crime rate if the law is routinely ignored without effective sanction (for example, widespread illegal strikes).

**Ethnic tensions:** According to the ICRG, this component is an assessment of the degree of tension within a country attributable to racial, nationality, or language divisions. Lower ratings are given to countries where racial and nationality tensions are high because opposing groups are intolerant and unwilling to compromise. Higher ratings are given to countries where tensions are minimal, even though such differences may still exist.

**Democratic Accountability:** According to the ICRG, this is a measure of how responsive government is to its people, on the basis that the less responsive it is, the more likely it is that the government will fall, peacefully in a democratic society, but possibly violently in a non-democratic one.

The points in this component are awarded on the basis of the type of governance enjoyed by the country in question. For this purpose, the ICRG defines the following types of governance.

Appendix A.2.1. Alternating Democracy

The essential features of an alternating democracy are:

- A government/executive that has not served more than two successive terms,
- Free and fair elections for the legislature and executive as determined by constitution or statute,
- The active presence of more than one political party and a viable opposition,
- Evidence of checks and balances among the three elements of government: executive, legislative and judicial,
- Evidence of an independent judiciary,
- Evidence of the protection of personal liberties through constitutional or other legal guarantees.

Appendix A.2.2. Dominated Democracy

The essential features of a dominated democracy are:

- Government/executive that has served more than two successive terms,
- Free and fair elections for the legislature and executive as determined by constitution or statute,
- The active presence of more than one political party,
- Evidence of checks and balances between the executive, legislature, and judiciary,
- Evidence of an independent judiciary,
- Evidence of the protection of personal liberties.

Appendix A.2.3. De Facto One-Party State

The essential features of a de facto one-party state are:

- A government/executive that has server more than two successive terms, or where the political/electoral system is designed or distorted to ensure the domination of governance by a particular government/executive,
- Holding of regular elections as determined by constitution or statute,
- Evidence of restrictions on the activity of non-government political parties (disproportionate media access between the governing and non-governing parties, harassment of the leaders and/or supporters of non-government political parties, the creation of impediments and obstacles affecting only the non-government political parties, electoral fraud, etc.).

Appendix A.2.4. De Jure One-Party State

The identifying feature of a one-party state is:

- A constitutional requirement that there be only one governing party,
- Lack of any legally recognized political opposition.

Appendix A.2.5. Autarchy

The identifying feature of an autarchy is: Leadership of the state by a group or single person, without being subject to any franchise, either through military might or inherited right. In an autarchy, the leadership might indulge in some quasi-democratic processes. In its most developed form this allows competing political parties and regular elections, through popular franchise, to an assembly with restricted legislative powers (approaching the category of a de jure or de facto one-party state). However, the defining feature is whether the leadership, i.e., the head of government, is subject to election in which political opponents are allowed to stand. In general, the highest number of risk points (lowest risk) is assigned to Alternating Democracies, while the lowest number of risk points (highest risk) is assigned to Autarchies.

**Bureaucracy quality:** According to the ICRG, the institutional strength and quality of the bureaucracy is another shock absorber that tends to minimize revisions of policy when governments change. Therefore, high points are given to countries where the bureaucracy has the strength and expertise to govern without drastic changes in policy or interruptions in government services. In these low-risk countries, the bureaucracy tends to be somewhat autonomous from political pressure and to have an established mechanism for recruitment and training. Countries that lack the cushioning effect of a strong bureaucracy receive low points because a change in government tends to be traumatic in terms of policy formulation and day-to-day administrative functions.

**Appendix B**

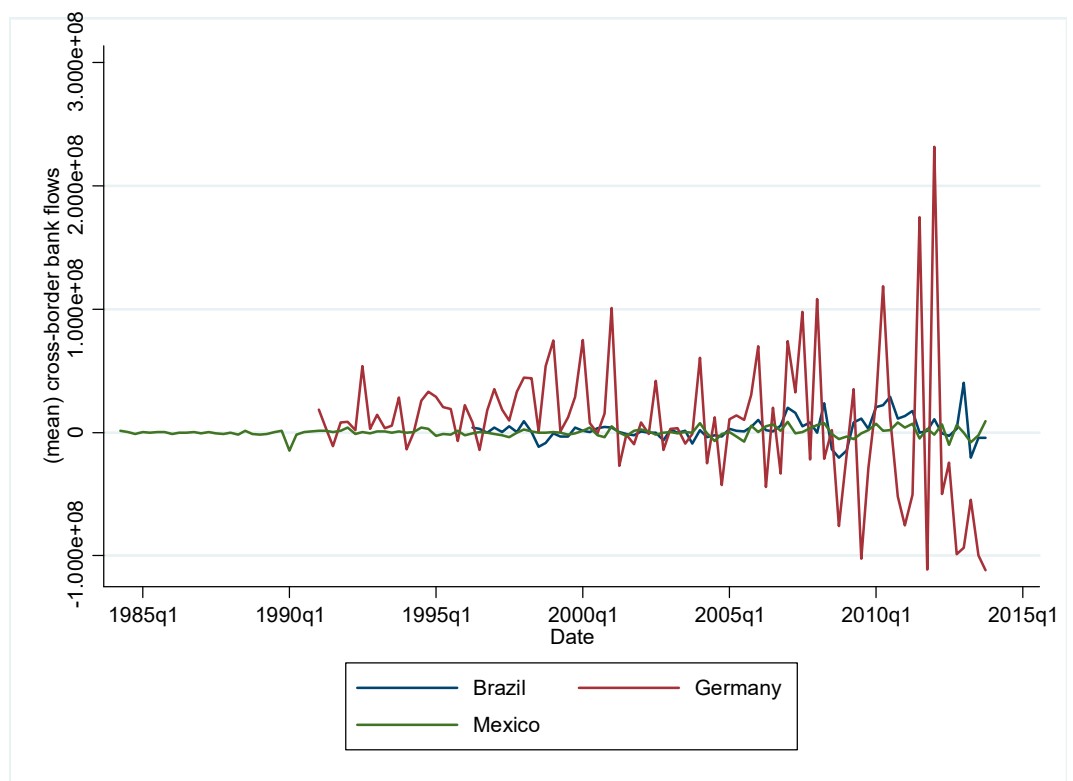

**Figure A1.** Cross-border bank flows: Germany, Brazil and Mexico.

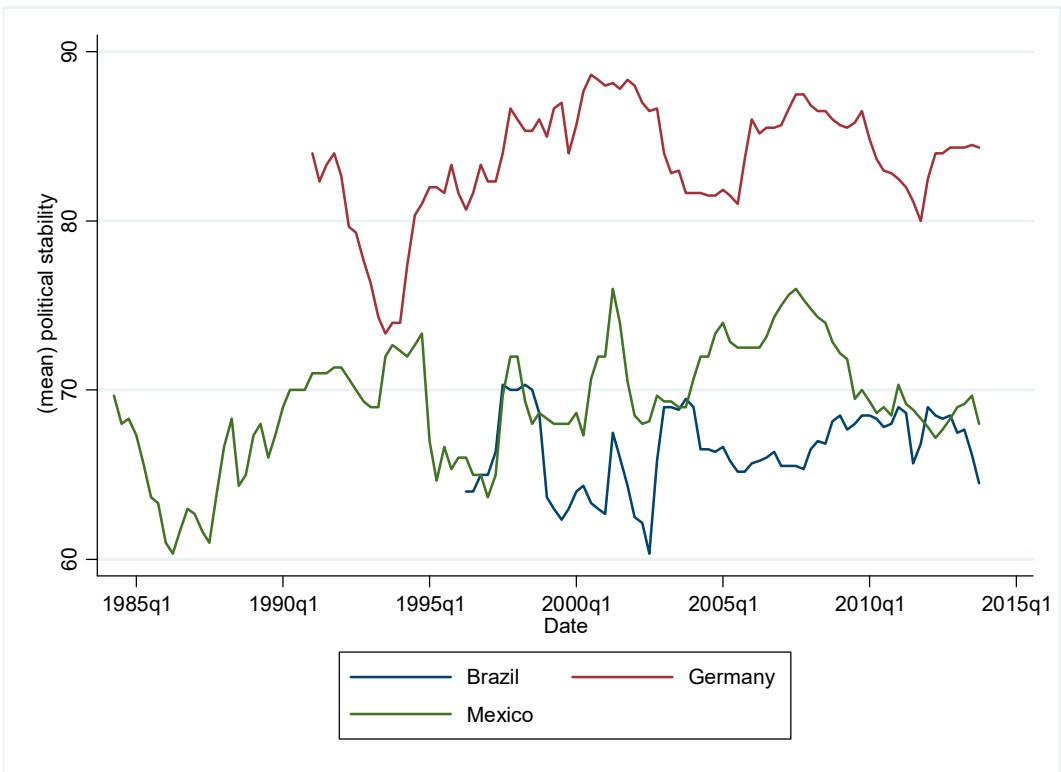

**Figure A2.** Political stability: Germany, Brazil and Mexico.

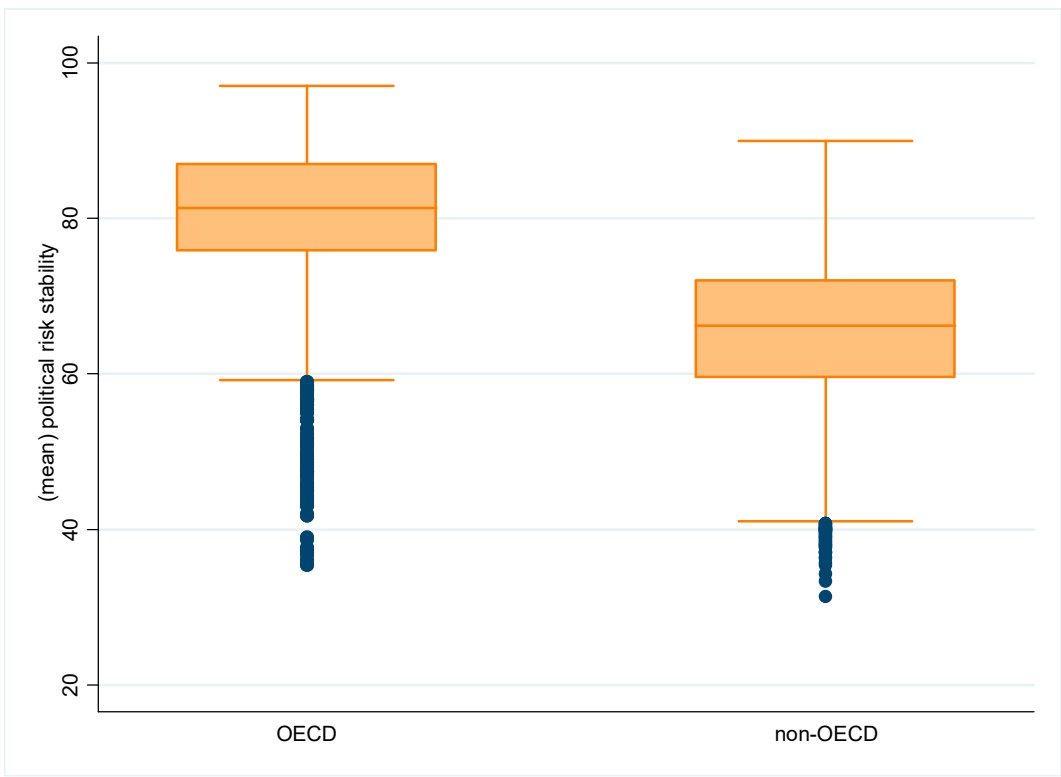

**Figure A3.** Box-plot: political stability (OECD vs. non-OECD countries).

**Table A1.** Descriptive statistics—all countries.

| Variables | N | Mean | Stand. Dev. | Min | Max |
|---|---|---|---|---|---|
| Bureaucracy quality | 5781 | 2.972 | 0.995 | 0 | 4 |
| Corruption | 5781 | 3.689 | 1.406 | 0 | 6 |
| Democratic accountability | 5781 | 4.969 | 1.235 | 1 | 6 |
| Economic risk | 5781 | 36.92 | 5.006 | 16.83 | 50 |
| Ethnic tensions | 5781 | 4.347 | 1.294 | 0 | 6 |
| Financial risk | 5781 | 38.80 | 5.927 | 11.33 | 50 |
| Government stability | 5781 | 8.040 | 1.794 | 1 | 12 |
| Investment profile | 5781 | 8.721 | 2.317 | 2 | 12 |
| Law and order | 5781 | 4.452 | 1.399 | 1 | 6 |
| Political stability | 5781 | 74.27 | 11.92 | 31.33 | 97 |
| Religious tensions | 5781 | 5.037 | 1.151 | 0.333 | 6 |
| Socioeconomic conditions | 5781 | 6.835 | 2.060 | 1.333 | 11 |
| GDP volume (% change) | 5781 | 3.271 | 4.154 | −20.93 | 24.31 |
| Cross-border bank flows | 5781 | $7.58 \times 10^{10}$ | $8.554 \times 10^{12}$ | $-3.394 \times 10^{14}$ | $4.409 \times 10^{14}$ |
| Logarithm cross-border bank flows | 5781 | 3.117 | 13.60 | −33.46 | 33.72 |

Notes: N is the number of observations. The sample includes all countries.

**Table A2.** Descriptive statistics—OECD countries.

| Variables | N | Mean | Stand. Dev. | Min | Max |
|---|---|---|---|---|---|
| Bureaucracy quality | 3415 | 3.513421 | 0.667745 | 1 | 4 |
| Corruption | 3415 | 4.369204 | 1.237542 | 2 | 6 |
| Democratic accountability | 3415 | 5.570473 | 0.7348982 | 2 | 6 |
| Economic risk | 3415 | 38.21798 | 4.509083 | 17.5 | 49.16667 |
| Ethnic tensions | 3415 | 4.538507 | 1.231432 | 1 | 6 |
| Financial risk | 3415 | 39.6429 | 5.841506 | 18.16667 | 50 |
| Government stability | 3415 | 8.000976 | 1.704215 | 2 | 12 |
| Investment profile | 3415 | 9.243924 | 2.30026 | 2.666667 | 12 |
| Law and order | 3415 | 5.18165 | 0.973454 | 1 | 6 |
| Political stability | 3415 | 80.06452 | 9.532913 | 35.33333 | 97 |
| Religious tensions | 3415 | 5.353245 | 0.8983487 | 1 | 6 |
| Socioeconomic conditions | 3415 | 7.689214 | 1.703555 | 2 | 11 |
| GDP volume (% change) | 3415 | 2.631638 | 3.534948 | −17.6413 | 16.7499 |
| Cross-border bank flows | 3415 | $1.28 \times 10^{11}$ | $1.11 \times 10^{13}$ | $-3.39 \times 10^{14}$ | $4.41 \times 10^{14}$ |
| Logarithm cross-border bank flows | 3415 | 4.06762 | 14.20838 | −33.45832 | 33.71973 |

Notes: N is the number of observations. The sample includes OECD countries.

**Table A3.** Descriptive statistics—non-OECD countries.

| Variables | N | Mean | Stand. Dev. | Min | Max |
|---|---|---|---|---|---|
| Bureaucracy quality | 2366 | 2.191251 | 0.8625091 | 0 | 4 |
| Corruption | 2366 | 2.706326 | 0.99162 | 0 | 6 |
| Democratic accountability | 2366 | 4.100944 | 1.292591 | 1 | 6 |
| Economic risk | 2366 | 35.0378 | 5.091587 | 16.83333 | 50 |
| Ethnic tensions | 2366 | 4.070442 | 1.332564 | 0 | 6 |
| Financial risk | 2366 | 37.58373 | 5.837498 | 11.33333 | 50 |
| Government stability | 2366 | 8.097281 | 1.915513 | 1 | 11.66667 |
| Investment profile | 2366 | 7.965272 | 2.124414 | 2 | 12 |
| Law and order | 2366 | 3.397718 | 1.238899 | 1 | 6 |
| Political stability | 2366 | 65.89598 | 9.862459 | 31.33333 | 90 |
| Religious tensions | 2366 | 4.580023 | 1.312078 | 0.333333 | 6 |
| Socioeconomic conditions | 2366 | 5.602494 | 1.90012 | 1.333333 | 11 |
| GDP volume (% change) | 2366 | 4.193731 | 4.762638 | −20.9328 | 24.3113 |
| Cross-border bank flows | 2366 | 467,646.5 | 4,409,704 | $-5.34 \times 10^{7}$ | $4.18 \times 10^{7}$ |
| Logarithm cross-border bank flows | 2366 | 1.745478 | 12.54209 | −17.79407 | 17.54779 |

Notes: N is the number of observations. The sample includes non-OECD countries.

**Table A4.** Correlation matrix.

| Variables | Log Cross-Border Bank Flows | Cross-Border Bank Flows | Political Stability | GDP Volume (% Change) | Economic Risk | Financial Risk | Bureaucracy Quality | Corruption | Democratic Accountability | Ethnic Tensions | Government Stability | Investment Profile | Law_and Order | Religious Tensions | Socioeconomic Conditions |
|---|---|---|---|---|---|---|---|---|---|---|---|---|---|---|---|
| Log cross-border bank flows | 1 | | | | | | | | | | | | | | |
| Cross-border bank flows | 0.0537 *** | 1 | | | | | | | | | | | | | |
| Political stability | 0.166 *** | 0.0202 | 1 | | | | | | | | | | | | |
| GDP volume (% change) | 0.188 *** | 0.0124 | −0.0962 *** | 1 | | | | | | | | | | | |
| Economic risk | 0.183 *** | 0.0107 | 0.626 *** | 0.160 *** | 1 | | | | | | | | | | |
| Financial risk | 0.164 *** | 0.0171 | 0.461 *** | 0.134 *** | 0.597 *** | 1 | | | | | | | | | |
| Bureaucracy quality | 0.101 *** | 0.0160 | 0.757 *** | −0.152 *** | 0.545 *** | 0.402 *** | 1 | | | | | | | | |
| Corruption | 0.0837 *** | −0.000420 | 0.684 *** | −0.120 *** | 0.433 *** | 0.342 *** | 0.743 *** | 1 | | | | | | | |
| Democratic accountability | 0.0787 *** | 0.00740 | 0.591 *** | −0.167 *** | 0.307 *** | 0.190 *** | 0.621 *** | 0.484 *** | 1 | | | | | | |
| Ethnic tensions | 0.0624 *** | 0.0212 | 0.509 *** | −0.0376 ** | 0.246 *** | 0.281 *** | 0.256 *** | 0.293 *** | 0.0720 *** | 1 | | | | | |
| Government stability | 0.117 *** | 0.0156 | 0.406 *** | 0.111 *** | 0.286 *** | 0.167 *** | 0.132 *** | 0.0990 *** | −0.0266 * | 0.152 *** | 1 | | | | |
| Investment profile | 0.136 *** | 0.0161 | 0.591 *** | −0.0265 * | 0.415 *** | 0.148 *** | 0.353 *** | 0.117 *** | 0.339 *** | 0.0655 *** | 0.380 *** | 1 | | | |
| Law and order | 0.121 *** | 0.0147 | 0.803 *** | −0.0975 *** | 0.528 *** | 0.437 *** | 0.708 *** | 0.701 *** | 0.492 *** | 0.409 *** | 0.211 *** | 0.277 *** | 1 | | |
| Religious tensions | 0.0555 *** | 0.0116 | 0.517 *** | −0.139 *** | 0.208 *** | 0.171 *** | 0.256 *** | 0.324 *** | 0.212 *** | 0.427 *** | 0.0813 *** | 0.177 *** | 0.303 *** | 1 | |
| Socioeconomic conditions | 0.146 *** | 0.0197 | 0.758 *** | −0.0636 *** | 0.590 *** | 0.343 *** | 0.656 *** | 0.550 *** | 0.441 *** | 0.197 *** | 0.216 *** | 0.555 *** | 0.611 *** | 0.251 *** | 1 |

$* \ p < 0.05$, $** \ p < 0.01$, $*** \ p < 0.001$.

**Table A5.** Panel estimates.

| Variables | (1) | (2) | (3) | (4) | (5) | (6) | (7) |
|---|---|---|---|---|---|---|---|
| Political stability $_{i,t-1}$ | 0.235 *** | 0.299 *** | 0.167 | 0.291 *** | 0.718 ** | 0.121 | 0.488 * |
| | (0.0771) | (0.0903) | (0.134) | (0.0846) | (0.269) | (0.117) | (0.245) |
| GDP vol. (% change) $_{i,t-1}$ | 0.385 *** | 0.293 *** | 0.366 *** | 0.181 | 0.448 ** | 0.214 ** | 0.589 *** |
| | (0.0566) | (0.103) | (0.0661) | (0.111) | (0.200) | (0.0829) | (0.150) |
| Economic risk $_{i,t-1}$ | 0.258 *** | 0.319 *** | −0.0551 | 0.123 | 0.0308 | 0.0671 | −0.159 |
| | (0.0936) | (0.103) | (0.125) | (0.142) | (0.202) | (0.139) | (0.239) |
| Financial risk $_{i,t-1}$ | 0.211 ** | 0.175 ** | 0.443 *** | 0.0152 | 0.481 * | 0.378 ** | 0.787 *** |
| | (0.0841) | (0.0781) | (0.148) | (0.0995) | (0.249) | (0.173) | (0.245) |
| Constant | −30.84 *** | −37.59 *** | −20.64 *** | −23.09 *** | −67.47 *** | −18.60 *** | −55.01 *** |
| | (5.815) | (7.833) | (6.580) | (8.326) | (24.16) | (6.033) | (17.17) |
| Observations | 5781 | 3415 | 2366 | 2599 | 816 | 1632 | 734 |
| R-squared | 0.152 | 0.198 | 0.182 | 0.113 | 0.157 | 0.182 | 0.180 |
| Number of countries | 71 | 34 | 37 | 34 | 34 | 35 | 35 |
| Country FE | YES | YES | YES | YES | YES | YES | YES |
| Time FE | YES | YES | YES | YES | YES | YES | YES |
| OECD | | YES | NO | YES | YES | NO | NO |
| Financial crisis | | | | BEFORE | AFTER | BEFORE | AFTER |

Robust standard errors in parentheses. *** $p < 0.01$, ** $p < 0.05$, * $p < 0.1$.

Table A6. Estimation results—components of political stability.

| Variables | (1) | (2) | (3) | (4) | (5) | (6) | (7) | (8) | (9) |
|---|---|---|---|---|---|---|---|---|---|
| Government stability$_{i, t-1}$ | 0.277 | | | | | | | | |
| | (0.251) | | | | | | | | |
| GDP vol. (% change)$_{i, t-1}$ | 0.307 *** | 0.278 ** | 0.275 ** | 0.323 *** | 0.329 *** | 0.326 *** | 0.321 *** | 0.316 *** | 0.328 *** |
| | (0.106) | (0.105) | (0.101) | (0.105) | (0.107) | (0.105) | (0.104) | (0.106) | (0.103) |
| Economic risk$_{i, t-1}$ | 0.421 *** | 0.355 *** | 0.341 *** | 0.400 *** | 0.406 *** | 0.435 *** | 0.428 *** | 0.437 *** | 0.413 *** |
| | (0.112) | (0.108) | (0.102) | (0.108) | (0.107) | (0.111) | (0.115) | (0.110) | (0.109) |
| Financial risk$_{i, t-1}$ | 0.268 *** | 0.272 *** | 0.236 *** | 0.289 *** | 0.269 *** | 0.264 *** | 0.285 *** | 0.293 *** | 0.227 *** |
| | (0.0730) | (0.0693) | (0.0719) | (0.0666) | (0.0690) | (0.0672) | (0.0678) | (0.0782) | (0.0772) |
| Socioeconomic conditions$_{i, t-1}$ | | 0.823 *** | | | | | | | |
| | | (0.272) | | | | | | | |
| Investment profile$_{i, t-1}$ | | | 1.011 *** | | | | | | |
| | | | (0.349) | | | | | | |
| Corruption$_{i, t-1}$ | | | | 1.369 ** | | | | | |
| | | | | (0.648) | | | | | |
| Religious tensions$_{i, t-1}$ | | | | | 1.536 *** | | | | |
| | | | | | (0.523) | | | | |
| Law and order$_{i, t-1}$ | | | | | | 0.645 | | | |
| | | | | | | (0.519) | | | |
| Ethnic tensions$_{i, t-1}$ | | | | | | | 0.926 * | | |
| | | | | | | | (0.477) | | |
| Democratic accountability$_{i, t-1}$ | | | | | | | | −0.135 | |
| | | | | | | | | (0.589) | |
| Bureaucracy quality$_{i, t-1}$ | | | | | | | | | 2.357 ** |
| | | | | | | | | | (1.132) |
| Constant | −23.09 *** | −24.85 *** | −24.87 *** | −27.56 *** | −28.48 *** | −24.32 *** | −26.21 *** | −21.43 *** | −26.87 *** |
| | (4.663) | (4.467) | (5.043) | (5.165) | (4.921) | (4.983) | (5.277) | (5.467) | (6.517) |
| Observations | 3415 | 3415 | 3415 | 3415 | 3415 | 3415 | 3415 | 3415 | 3415 |
| R-squared | 0.192 | 0.195 | 0.197 | 0.194 | 0.194 | 0.192 | 0.193 | 0.192 | 0.194 |
| Number of countries | 34 | 34 | 34 | 34 | 34 | 34 | 34 | 34 | 34 |
| Country FE | YES | YES | YES | YES | YES | YES | YES | YES | YES |
| Time FE | YES | YES | YES | YES | YES | YES | YES | YES | YES |
| OECD | YES | YES | YES | YES | YES | YES | YES | YES | YES |

Notes: The dependent variable is the logarithm of the cross-border bank flows from all countries to country $i$ in quarter $t$. Robust standard errors in parentheses. *** $p < 0.01$, ** $p < 0.05$, * $p < 0.1$.

**Table A7.** Estimation results for OECD countries—components of political stability (before financial crisis).

| Variables | (1) | (2) | (3) | (4) | (5) | (6) | (7) | (8) | (9) |
|---|---|---|---|---|---|---|---|---|---|
| Government stability $_{i,\,t-1}$ | 0.272 | | | | | | | | |
| | (0.273) | | | | | | | | |
| GDP volume (% change) $_{i,\,t-1}$ | 0.209 * | 0.182 | 0.189 | 0.236 ** | 0.218 * | 0.222 * | 0.222 * | 0.225 * | 0.222 * |
| | (0.112) | (0.118) | (0.113) | (0.112) | (0.117) | (0.115) | (0.113) | (0.116) | (0.114) |
| Economic risk $_{i,\,t-1}$ | 0.239 | 0.202 | 0.191 | 0.218 | 0.235 | 0.252 | 0.248 | 0.235 | 0.220 |
| | (0.172) | (0.165) | (0.142) | (0.152) | (0.157) | (0.167) | (0.175) | (0.150) | (0.166) |
| Financial risk $_{i,\,t-1}$ | 0.130 | 0.136 | 0.103 | 0.154 * | 0.135 | 0.119 | 0.149 * | 0.128 | 0.0895 |
| | (0.0912) | (0.0848) | (0.0973) | (0.0839) | (0.0854) | (0.0826) | (0.0823) | (0.0991) | (0.0926) |
| Socioeconomic conditions $_{i,\,t-1}$ | | 0.592 * | | | | | | | |
| | | (0.318) | | | | | | | |
| Investment profile $_{i,\,t-1}$ | | | 0.738 * | | | | | | |
| | | | (0.376) | | | | | | |
| Corruption $_{i,\,t-1}$ | | | | 0.998 | | | | | |
| | | | | (0.636) | | | | | |
| Religious tensions $_{i,\,t-1}$ | | | | | 1.093 | | | | |
| | | | | | (0.690) | | | | |
| Law and order $_{i,\,t-1}$ | | | | | | 0.843 | | | |
| | | | | | | (0.512) | | | |
| Ethnic tensions $_{i,\,t-1}$ | | | | | | | 0.919 | | |
| | | | | | | | (0.649) | | |
| Democratic accountability $_{i,\,t-1}$ | | | | | | | | 0.442 | |
| | | | | | | | | (0.738) | |
| Bureaucracy quality $_{i,\,t-1}$ | | | | | | | | | 2.297 |
| | | | | | | | | | (1.395) |
| Constant | −10.56 * | −11.66 * | −11.52 * | −13.46 * | −14.13 ** | −12.57 ** | −13.87 ** | −10.41 | −13.85 * |
| | (5.904) | (5.797) | (6.197) | (6.730) | (6.907) | (6.092) | (6.381) | (6.859) | (7.796) |
| Observations | 2599 | 2599 | 2599 | 2599 | 2599 | 2599 | 2599 | 2599 | 2599 |
| R-squared | 0.105 | 0.107 | 0.108 | 0.107 | 0.106 | 0.106 | 0.106 | 0.105 | 0.107 |
| Number of countries | 34 | 34 | 34 | 34 | 34 | 34 | 34 | 34 | 34 |
| Country FE | YES | YES | YES | YES | YES | YES | YES | YES | YES |
| Time FE | YES | YES | YES | YES | YES | YES | YES | YES | YES |
| Financial crisis | BEFORE | BEFORE | BEFORE | BEFORE | BEFORE | BEFORE | BEFORE | BEFORE | BEFORE |

Robust standard errors in parentheses. *** $p < 0.01$, ** $p < 0.05$, * $p < 0.1$.

**Table A8.** Estimation results for OECD countries—components of political stability (after financial crisis).

| Variables | (1) | (2) | (3) | (4) | (5) | (6) | (7) | (8) | (9) |
|---|---|---|---|---|---|---|---|---|---|
| Government stability $_{i,\,t-1}$ | 0.450 | | | | | | | | |
| | (0.606) | | | | | | | | |
| GDP volume (% change) $_{i,\,t-1}$ | 0.509 ** | 0.470 ** | 0.423 ** | 0.482 ** | 0.531 *** | 0.504 ** | 0.523 ** | 0.522 ** | 0.522 ** |
| | (0.199) | (0.188) | (0.194) | (0.193) | (0.190) | (0.189) | (0.198) | (0.192) | (0.192) |
| Economic risk $_{i,\,t-1}$ | 0.128 | 0.0385 | −0.0299 | 0.0867 | 0.149 | 0.121 | 0.126 | 0.128 | 0.126 |
| | (0.228) | (0.221) | (0.199) | (0.210) | (0.231) | (0.226) | (0.230) | (0.225) | (0.228) |
| Financial risk $_{i,\,t-1}$ | 0.580 ** | 0.553 ** | 0.381 | 0.749 *** | 0.614 ** | 0.574 ** | 0.591 ** | 0.596 ** | 0.591 ** |
| | (0.252) | (0.236) | (0.245) | (0.253) | (0.252) | (0.248) | (0.250) | (0.254) | (0.250) |
| Socioeconomic conditions $_{i,\,t-1}$ | | 2.198 ** | | | | | | | |
| | | (0.937) | | | | | | | |
| Investment profile $_{i,\,t-1}$ | | | 1.701 ** | | | | | | |
| | | | (0.661) | | | | | | |
| Corruption $_{i,\,t-1}$ | | | | 8.032 *** | | | | | |
| | | | | (1.767) | | | | | |
| Religious tensions $_{i,\,t-1}$ | | | | | 4.677 *** | | | | |
| | | | | | (1.472) | | | | |
| Law and order $_{i,\,t-1}$ | | | | | | −3.458 | | | |
| | | | | | | (2.277) | | | |
| Ethnic tensions $_{i,\,t-1}$ | | | | | | | 0.124 | | |
| | | | | | | | (7.436) | | |
| Democratic accountability $_{i,\,t-1}$ | | | | | | | | 1.235 | |
| | | | | | | | | (4.294) | |
| Bureaucracy quality $_{i,\,t-1}$ | | | | | | | | | - |
| Constant | −20.36 | −31.00 ** | −21.91 * | −52.48 *** | −43.55 ** | 1.400 | −17.66 | −24.55 | −17.13 |
| | (13.11) | (13.46) | (12.30) | (15.87) | (17.54) | (18.87) | (34.64) | (27.66) | (12.50) |
| Observations | 816 | 816 | 816 | 816 | 816 | 816 | 816 | 816 | 816 |
| R-squared | 0.149 | 0.155 | 0.156 | 0.162 | 0.149 | 0.149 | 0.148 | 0.148 | 0.148 |
| Number of countries | 34 | 34 | 34 | 34 | 34 | 34 | 34 | 34 | 34 |
| Country FE | YES | YES | YES | YES | YES | YES | YES | YES | YES |
| Time FE | YES | YES | YES | YES | YES | YES | YES | YES | YES |
| Financial crisis | AFTER | AFTER | AFTER | AFTER | AFTER | AFTER | AFTER | AFTER | AFTER |

Robust standard errors in parentheses. *** $p < 0.01$, ** $p < 0.05$, * $p < 0.1$.

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
