# Peer review of "Political Stability and Bank Flows: New Evidence"

_jrfm, doi:10.3390/jrfm13030056_

Round 1
Reviewer 1 Report
I believe the paper would benefit by investigating whether/how the results depend on a financial crisis. Given that the sample includes the financial crisis in 2008, it is possible to elaborate on an event-based approach as well as pre-/post-sample analysis.
Reviewer 2 Report
The paper presents an empirical analysis of cross-border banking flows among a large set of (OECD and non-OECD) countries that can usefully complement the existing literature on this topic. In fact, the focus of the paper is on several dimensions of political risk as factors affecting the magnitude and dynamics of international bank flows, whereas the literature has analyzed the effect of other (mainly economic and financial) sources of risk on bank flows.
The data set is valuable, comprising around 70 countries with a large variability of the amount of political risk, carefully collected and described in the Appendix.
Some improvements are still possible and desirable, especially in terms of presentation of descriptive statistics and results:
1) in Table 1, cross-border bank flows should be reported in levels and not logs (as in the regressions) for easier interpretability; moreover, separate descriptive statistics for OECD and non-OECD countries could be useful to the reader.
2) In the last paragraph of section 3.1 the following sentence is not clear: "However, this result may be unbiased considering the fact that other factors may be capturing part of the effect of the political stability" (p. 6).
3) When (Section 3.2) components of the political stability measure are introduced and separately considered, some information on the correlation among them could be provided (as in Table 2 for the political, economic and financial risk overall measures), to assess whether in fact they manage to capture different aspects of the phenomenon investigated.
4) (related to the previous comment) The choice of including in the estimated regression of Table 4 one component of the political stability measure at a time should be at least discussed and motivated.
The English language still needs some revision, in order to avoid repeated mistakes (e.g. at least 5 can be counted in the opening paragraph of the introduction) that make it difficult to enjoy reading the paper. Moreover, some over-confident statements should perhaps be avoided (e.g. the last sentence of subsection 2.1.1. on p. 4). Finally, several references should be checked and updated (e.g. the Choi-Furceri paper has been published in 2019 in the Journal of International Money and Finance).
Reviewer 3 Report
Major issue:
1) The factors of the analytic models are not well supported by the introduction. For instance, the author discussed about the global risk aversion and financial/economic uncertainty in the introduction section. However, there is not a specific factors named global risk aversion. Instead, there are similar variables: economic risk, and financial risk. However, risk aversion is different concept with risk. In addition, there should be additional explanation about the differences between economic risk and financial risk as the conceptual/theoretical background. Furthermore, what is the socioeconomic condition? This was just listed as the factors but not explained conceptually or theoretically. To solve this, all factors in Table 3 and 4 should be well explained and introduced in the introduction part or theoretical background part.
2) In the methodology part, there is only one function to describe the main analytic model. However, Table 3 and 4 showed multiple estimations. The author should specify all models in the paragraph. Even though only one main function well describe all models, the paragraph should indicates how many actual models will be analyzed.
3) Conclusion does not contain any critical finding that make the manuscript be unique. In other words, the conclusion seems to be very common knowledge that we have already known. In addition, the conclusion does not contain how the findings are important to interpret the global situation. To solve this issue, the author may indicate (a) the uniqueness of the manuscript and (b) the practical implication from the findings.
Minor issue:
1) There are too many paragraphs that were made by one-sentence.
Round 2
Reviewer 1 Report
The revised version of the paper includes a number of improvements and makes an overall better contribution than the former version.
Reviewer 3 Report
Even though the authors mentioned that they explained the main drivers of international bank business in the introduction section, it is not well recognizable. Specifically, there are lacks of why those drivers should be considered and chosen for the research. However, it is minor issue because the importance of theoretical/conceptual support can be differently weighted by fields.
